# Secondary care consultant clinicians' experiences of conducting emergency care and treatment planning conversations in England: an interview-based analysis

Karin Eli [1], Cynthia Ochieng [2], Claire Hawkes [1], Gavin D Perkins [1,3], Keith Couper [1,3], Frances Griffiths [1], Anne-Marie Slowther [1]

FG and A-MS are joint senior authors.

¹Warwick Medical School, University of Warwick, Coventry, UK
²School of Medicine, Cardiff University, Cardiff, UK
³Critical Care Unit, University Hospitals Birmingham NHS Foundation Trust, Birmingham, UK

**Correspondence to**
Dr Karin Eli;
Karin.Eli@warwick.ac.uk

## ABSTRACT

**Objective** To examine secondary care consultant clinicians' experiences of conducting conversations about treatment escalation with patients and their relatives, using the Recommended Summary Plan for Emergency Care and Treatment (ReSPECT) process.

**Design** Semi-structured interviews following ward round observations.

**Setting** Two National Health Service hospitals in England.

**Participants** Fifteen medical and surgical consultants from 10 specialties, observed in 14 wards.

**Analysis** Interview transcripts were analysed using thematic analysis.

**Results** Three themes were developed: (1) determining when and with whom to conduct a ReSPECT conversation; (2) framing the ReSPECT conversation to manage emotions and relationships and (3) reaching ReSPECT decisions. The results showed that when timing ReSPECT conversations, consultant clinicians rely on their predictions of a patient's short-term prognosis; when framing ReSPECT conversations, consultant clinicians seek to minimise distress and maximise rapport and when involving a patient or a patient's relatives in decision-making discussions, consultant clinicians are guided by their level of certainty about the patient's illness trajectory.

**Conclusions** The management of uncertainty about prognoses and about patients' emotional reactions is central to secondary care consultant clinicians' experiences of timing and conducting ReSPECT conversations.

## Strengths and limitations of this study

► Data were collected from consultant clinicians from 10 specialties, thereby representing diverse secondary care environments and clinical attitudes to emergency care and treatment planning.
► Each clinician was first shadowed during a ward round and then interviewed, thus grounding the interviews in specific and varied case examples.
► The findings reported in this paper are limited by the study's focus on consultants' interviews; as other members of multidisciplinary teams also participate in the Recommended Summary Plan for Emergency Care and Treatment (ReSPECT) conversations, including their perspectives and experiences would have been valuable.
► The interviews took place within the first year of ReSPECT implementation in the two study sites, such that some findings may reflect experiences related to early implementation.

## INTRODUCTION

UK clinical practice guidelines indicate that cardiopulmonary resuscitation (CPR) may be withheld when clinicians predict it would not succeed, if the patient refuses CPR or following careful clinical assessment of the benefits and burdens of CPR.[1 2] While Do Not Attempt Cardiopulmonary Resuscitation (DNACPR) guidelines are clearly articulated, several studies and reviews have found that, in practice, DNACPR processes are fraught with ambiguity. Clinicians have varying, sometimes divergent understandings of DNACPR decision-making processes, leading to inconsistencies in how decisions are made, implemented and recorded.[3–6] These inconsistencies may lead to lower quality of care; indeed, some clinicians misinterpret DNACPR decisions as limiting other aspects of treatment, while others administer CPR inappropriately, failing to follow patients' wishes for the withholding of resuscitation.[3 7–10] Notably, clinicians often communicate poorly about DNACPR with patients and their relatives, and some are reluctant to discuss resuscitation, thereby excluding patients from the decision-making process.[4 6 7 11–13]

This paper is part of a larger study, funded by the National Institute for Health Research, which evaluates the Recommended Summary

Plan for Emergency Care and Treatment (ReSPECT).[14] Launched in 2017 across NHS Trusts, ReSPECT is an emergency care treatment plan (ECTP) developed in response to the gaps observed in the DNACPR process. ReSPECT builds on research conducted in the United States, the UK and Canada, which found that programmes that integrate DNACPR with discussions about wider goals of treatment increase clarity about trajectories of care and reduce harm to patients.[15] As an ECTP which records clinical recommendations that take into account patients' values and preferences, ReSPECT places resuscitation within a wider context of treatments that should or should not be considered in an emergency situation.

The authors of ReSPECT emphasise that it is a process designed to guide clinicians in discussing with patients what might be optimal treatment choices for them, with the ReSPECT form acting as a prompt and summary record of the discussion and its outcomes.[16] The form and its associated guidance documents were developed in 2016 by the ReSPECT working group. Chaired by the Resuscitation Council (UK) and Royal College of Nursing, the ReSPECT working group had representation from patients, professional organisations (Royal Colleges, British Medical Association) regulatory bodies (General Medical Council, Nursing Midwifery Council), the Care Quality Commission, and National Health Service (NHS) organisations (Acute, Community and Ambulance Trusts). The completed ReSPECT form is held by the patient, allowing them to communicate their treatment plans when they move from one healthcare setting to another.

In the present paper, we report findings from interviews with secondary care consultant clinicians (henceforth, consultants) in two NHS organisations that had recently implemented ReSPECT, exploring why, when and with whom they choose to have ReSPECT conversations. Our aim is to inform the future development of the process and the current implementation of ReSPECT across the NHS and to provide focus to further qualitative research on how ReSPECT becomes integrated into health professionals' practice.

## METHODS

Fifteen consultants (six women, nine men) from two acute NHS teaching hospitals in England were interviewed from August to December 2017 as part of a wider ongoing study, aimed at evaluating the implementation of the ReSPECT process. The 15 consultants represented 14 wards and 10 specialties. In site 1, interviews took place 7–10 months after ReSPECT had been implemented, and in site 2, 11–12 months after implementation. We observed no differences related to ReSPECT implementation timelines between the two sites.

Potential participants were identified by the local principal investigator at each of the participating hospitals through purposive sampling designed to represent a range of views about the ReSPECT process, as well as a diversity of clinical areas that could be replicated across NHS trusts (three medical specialties, a surgical specialty and orthopaedics). The local PIs or research nurses asked for volunteer participants from these specialties and the study's research fellow scheduled ward round observations directly with the participating consultants, to ensure that observations did not place an undue burden on their clinical practice. All participating consultants provided written informed consent prior to taking part in the study.

The research fellow, a public health researcher, shadowed each consultant during a ward round, to observe when and how consultants engaged in ReSPECT conversations with their patients. Shadowing is a structured observation technique,[17] which has been identified as appropriate for qualitative research on clinicians' experiences and practices.[18 19] To ensure that patients, relatives and staff were aware that observations were taking place, study posters were displayed in the selected wards, and the research fellow wore a scrubs uniform top with the word 'researcher' printed clearly on both the front and the back. During each shadowing period, the participating consultant introduced the researcher to each patient (and the patient's relatives, if present) and informed them that they could request that the researcher leave if they wished. A brief information leaflet was left with the patient. The researcher interviewed each consultant following the observation, typically within 24–48 hours. The interviews were semi-structured and were designed to explore each consultant's decision-making about holding a ReSPECT conversation in three observed cases, as well as the consultant's wider experiences with ReSPECT. If the researcher observed three to five ReSPECT conversations (five was the maximum she had observed in any of her observation sessions), she aimed to select at least two of these cases for discussion during the interviews. In the majority of interviews, the researcher also selected one or more cases where she thought a ReSPECT conversation might have been appropriate, to explore with the clinician why they chose not to hold a ReSPECT conversation in those cases. The interview topic areas were developed based on the study's research questions and the literature, and the observation and interview approach was checked with members of the study team with relevant clinical experience. The interviews lasted from 15 to 53 min, with a median time of 37 min, and were digitally recorded and transcribed.

Interview transcripts were analysed by the study's senior research fellow (SRF), a medical anthropologist, using thematic analysis.[20] First, the SRF read the interview transcripts to identify initial codes. The transcripts were then coded closely, with most codes developed at the level of sentences or sentence clauses. The SRF reviewed the coded interviews, and grouped the codes to develop themes. The themes were continuously revised throughout the process of reviewing the coded interviews, leading to 16 emerging themes, which were grouped into overarching themes. To ensure intercoder reliability, 4 of the 15 interviews were analysed independently by another SRF, a health services

researcher. The two SRFs discussed the codes, identified differences and potential disagreements and discussed these until they reached consensus. In total, five initial overarching themes were developed: three focused on the ReSPECT conversation, one focused on consultants' value judgements and one focused on the ReSPECT form. After they achieved consensus, the SRFs discussed the five overarching themes the two senior coauthors, doctors with research expertise in medical ethics and medical sociology. Together, they decided to focus the analysis on the three overarching themes concerned with the ReSPECT conversation, as these themes most closely responded to the study's aim of exploring why, when and with whom consultants choose to have ReSPECT conversations. Finally, the first SRF reviewed all interview transcripts to ensure the three themes represented the data accurately. Throughout the analytic process, coding was conducted using word processing software. To maintain participant confidentiality, the gender-neutral pronouns they/them are used throughout the manuscript to refer to all consultants.

## Patient and public involvement

The study is supported by a patient and public involvement (PPI) group, which informed the study design and the development of models of recruitment and consent. Additionally, PPI group members provided feedback on a draft of the manuscript. They agreed with the manuscript's findings and offered suggestions for areas to investigate further in our future research, in light of their own experiences as patients or carers.

## RESULTS
### Theme 1: determining when and with whom to conduct a ReSPECT conversation: uncertainty management and catalysts for discussion

Given time constraints, consultants had to determine which patients were most in need of a ReSPECT conversation, and when this conversation should be conducted. Making these determinations was fraught with uncertainty. To manage this uncertainty, consultants relied on their predictions and imaginings of patients' immediate futures, using the ward round to piece together prognostic puzzles. For example, explaining why they did not initiate a ReSPECT conversation with a patient in her 90s, this consultant said:

> she's otherwise recovering well (…) I thought the chances of her, as it were, needing any of the interventions you might discuss on a ReSPECT form were small. (Site 1, C04)

The key to initiating a ReSPECT conversation, this consultant later explained, was predicting a trajectory of deterioration:

> It's the deteriorating patients, patients with end-of-life conditions for whatever reason, be it cancer or

organ failure and any patient where they might suddenly deteriorate. (Site 1, C04)

Another consultant said they used the ward round to predict whether a patient was likely to experience a 'catastrophe'; such a prediction, they said, would warrant initiating a ReSPECT conversation:

> Particularly where you're seeing acutely ill patients and you're seeing them for the first time as, as an acute physician, I think the prompt is how likely you think it is that this patient may have a catastrophe, may have a cardiac arrest, may have a sudden severe deterioration. (Site 1, C07)

This consultant, like numerous others in the sample, linked the focus on predicted deterioration to the time constraints of the ward round. As another consultant explained:

> I think in the context of a post-take ward round where I am time limited I prioritise those patients for whom these conversations are most likely to be required for this admission. So it may well be that ReSPECT conversations were appropriate for more of the patients that I saw today in terms of potentially being last year of life. But they were not decisions that were required today. (Site 2, C04)

While time constraints were frequently cited, they were not the only factor underlying consultants' focus on predicted deterioration. Explaining why they were less likely to initiate a ReSPECT conversation with some patients, this consultant positioned their decision-making as culturally embedded:

> I think for the time being the culture is still the ReSPECT form is mainly for when people deteriorate. I think sometimes asking people a hypothetical question when they're really quite well, it's difficult to frame it. (Site 1, C06)

This consultant linked their focus on deterioration as the primary prompt for a ReSPECT conversation to the difficulty of asking patients to imagine a hypothetical difficult scenario. Imagining difficult scenarios, however, was central to ReSPECT conversations. Since initiating ReSPECT conversations depended on clinicians' predictions of patients' short-term prognoses, ReSPECT conversations engaged patients with clinicians' predictive thinking. This process was often challenging, as patients did not necessarily share in the logics and concepts of time posed by their clinicians:

> [P]eople find, 'What if?' challenging. So if I say, 'What if you're going to deteriorate? We need to make a decision what we would do about ITU'. A lot of patients and relatives will hear about us saying, 'You're deteriorating, you're going to need ITU'. They don't hear the 'What if?' (Site 2, C04)

Like others, this consultant explained that while they framed possible future scenarios in subjunctive—that is, potential or conditional—terms, patients and relatives tended to understand these in definitive future terms. Another consultant explained that, rather than joining a dialogue on potential scenarios, some patients and relatives expressed distress over what they understood as a terminal prognosis:

…even though I've said 'I am going to let you go home now, have you thought about what you would like in the future?', and then they say 'why am I going to die? You're telling me I am going to die aren't you!' (Site 1, C08)

In addition to predicted deterioration, consultants timed ReSPECT conversations according to calculations of risk related to a wider network of actors. The timing of ReSPECT conversations therefore implicated patients and other clinicians. For example, some consultants said they were reluctant to hold ReSPECT conversations with preoperative patients, as these discussions could bias surgeons or demoralise patients:

(The patient) was going to undergo an operation and I feel very uncomfortable discussing resuscitation just before the operation (…) if he does not want to be resuscitated, it influences the surgeon to some extent. (Site 2, C01)

…we don't often talk about ReSPECT form because it's, you know, when you deteriorate. And in some ways with elective surgery they're very much focused on consenting for surgery and talking about rehab after surgery rather than deterioration from surgery. (Site 1, C02)

Elsewhere in the interview, this consultant said the best time to initiate a ReSPECT conversation was immediately after surgery—a particularly opportune time because, while risk of complications was higher after surgery, patients' distress was likely to be lower, and relatives were likely to be present:

…often, often there's a family around at that point because it's usually an emergency admission and the family come in a day or two later. So you can involve the parties that you need to at that point in time. It's a relatively rare event for a patient to die on the operating table, if they're going to, if they're going to succumb it's usually over the following few days. (Site 1, C02)

The presence of relatives was central to the timing of many ReSPECT conversations, not least because conducting the ReSPECT process with patients who lack capacity requires the participation of an individual close to the patient. In this example, a consultant discussed a case where a visit from a patient's relatives prompted a ReSPECT conversation:

[H]is family were there so I took the opportunity while they were all there to express not only that he was perhaps more unwell than they had recognised, and that he was getting better with treatment, and to explore what their feelings were about escalation of care, particularly whether intensive care would be appropriate for him. (Site 2, C04)

This consultant considered the presence of relatives crucial in timing the ReSPECT conversation due to uncertainty about escalating the patient's care. Other consultants, however, spoke of the presence of relatives as important for finalising and communicating a medical decision, rather than deliberating about a trajectory of care.

### Theme 2: conducting the ReSPECT conversation: managing emotions and relationships

Most ReSPECT conversations implicated a triad of patient, clinician(s) and relative(s). For patients who lacked capacity, this triad was essential to the ReSPECT process, with relatives or other advocates called on to speak on the patient's behalf. However, while patients with capacity could speak privately with their clinicians, they often involved their relatives, framing the ReSPECT conversation and their own decision-making as familial. In these cases, consultants clarified they included relatives in the conversation, but did not involve them in decision-making:

…you've had a discussion, the patient says, 'Look, I don't want anything done, doctor', I think it's still very valuable to the next of kin to know that. (…) but we're not asking the family to participate in the discussion if the patient has already made their wishes clear in a reliable way. (Site 1, C07)

Although consultants tended to describe relatives' involvement as valuable, they also described it as potentially problematic, characterising family members as either compliant and 'sensible', or as non-compliant and 'difficult'. According to some consultants, relatives sometimes challenged clinical decisions—specifically, decisions against resuscitation—because they misunderstood what 'not for resuscitation' meant for the patient's future care:

Sometimes you have relatives who are very emotional, sometimes they think when you say 'not for resuscitation' means you're going to stop all treatment. (Site 2, C03)

In other cases, consultants said relatives misunderstood their role in the ReSPECT conversation as that of 'decision-maker', worrying about how a 'not for resuscitation' decision might reflect on them:

Often what happens is the relatives feel that you're asking them to make the decision… and again because they've been misled by the media, they feel that

if they say, yes, make them not for resuscitation, that they might be seen as a money grabbing. (Site 2, C02)

Disagreement between clinicians and patients' relatives could carry consequences for patient care, particularly if relatives who held lasting power of attorney (LPA) attempted to overturn a clinical decision. In those cases, consultants advocated for their clinical decision, taking the role of acting on the patient's behalf:

I try to explain to them that by keeping them alive, you are, you are, you are prolonging their agony. (…) I try to avoid confrontation with them (…) But sometimes we have to, when I can see clearly that there is going to be harm, then I have to, even if they have the LPA. (Site 2, C01)

Another relational aspect of the ReSPECT conversation was the consideration of other, sometimes absent, clinicians. Several consultants spoke about the importance of identifying the 'right' clinician to conduct a ReSPECT conversation—often, the consultant or the general practitioner (GP) regularly charged with the patient's care. In post-take ward rounds, some consultants avoided conducting ReSPECT conversations with patients who were usually seen by their colleagues. In this example, a consultant explained why they chose not to complete a ReSPECT form with a patient who had a localised infection:

I could've completed a ReSPECT form but I didn't because I, effectively I'm not looking at her (as her) responsible consultant. (Site 2, C02)

Later in the interview, this consultant explained that the patient's condition did not warrant an urgent ReSPECT conversation. Given the lack of urgency, they deferred to the patient's 'usual physicians, who obviously know her prognosis'. It would be inappropriate, this consultant argued, to conduct a ReSPECT conversation with a patient whose consultants evidently did not deem it necessary.

Consultants deferred ReSPECT conversations until the 'right' clinicians could conduct them because patients' usual consultants were more knowledgeable about these patients' medical histories, and because these usual consultants had established rapport with the patients. In this example, a consultant explained why they chose to conduct a ReSPECT conversation but leave the final decision for a future discussion between the patient and her usual consultant. The patient, this consultant explained, was not 'receptive' to an earlier ReSPECT conversation with her usual consultant. As such, they viewed their role as providing a second opinion to support the consultant's, rather than as finalising a ReSPECT decision.

…I didn't feel as if I was going to be welcomed to take that further with her myself. So I thought it was better than to say, to see her back to her normal consultant then the next time. (Site 2, C05)

For similar reasons, other consultants suggested that ReSPECT conversations were best conducted in primary care settings, led by patients' GPs rather than by clinicians they first met during an acute care admission. In response to the researcher's question, 'So you think this is something that should be discussed in the community?', this consultant said:

Definitely because I think it makes… patients feel less vulnerable… when they are in hospital they feel vulnerable plus they don't know us (…) they might have known the GP or have some sort of on-going or community matron or something that's a bit more of a long term relationship. (Site 1, C03)

The importance consultants placed on rapport was closely connected to their concerns over trustworthiness. Worries about being perceived as untrustworthy led some consultants to avoid or delay ReSPECT conversations with some patients. As described by consultants, ReSPECT conversations, if not framed carefully, could undermine the process of building trust with patients.

One of my worries is that patients, if you're not careful with your language, a patient might interpret a discussion about what to do in the event of deterioration, escalation, CPR, etcetera, as you giving up on them, as you not being prepared to do everything that you can to get them over their illness. (Site 1, C07)

The timing of ReSPECT conversations could also affect trust building. This consultant, for example, suggested that initiating a ReSPECT conversation too early would erode the patient's trust:

… you want to make sure you still have the rapport with the patient, that they see you as somebody that's there to help them (…) and if you feel that the patient is not quite ready to talk about it or they don't want to then if you kind of push it they'll see you negatively. (Site 1, C03)

Another consultant described a case where they conducted a ReSPECT conversation before a seriously ill patient underwent surgery. While the ReSPECT conversation was carefully timed from a medical perspective, it forced the patient to confront difficult scenarios that destabilised her trust in the surgeon:

So she doesn't want to talk about whether or not she's going to die on the operating table, or whether or not she's going to get her post-op chest infection or a lung embolus or whatever else could happen. But that process makes us talk about it at that point in time. (…) as soon as you mention that sentence about what would you like to do and if things were taking a turn for the worse (…) she's switched, she's completely switched off. (Site 1, C02)

Notably, consultants were concerned about being perceived as trustworthy because they identified the

ReSPECT conversation as a catalyst for potential distress for patients and relatives. To manage the difficult emotions that often arose during ReSPECT conversations, consultants used various techniques: from avoiding the conversation if the patient was expected to react aggressively or become overwhelmed, to initiating a series of conversations to ease patients and relatives into their future trajectory. For example, one consultant deferred ReSPECT conversations with patients recently diagnosed with terminal cancer to avoid overwhelming them:

> I've generally just told them they've got incurable cancer and it, to go on straight from that to a ReSPECT conversation is too much. But I will say that it exists and that it may be something they want to consider and then ask somebody else to follow it up. (Site 2, C04)

Alongside concerns over patients' emotional wellbeing, several consultants said that previous experiences with patients or relatives who became upset made them cautious about initiating and framing ReSPECT conversations. One consultant, who explained that 'we worry about the angry and anxious one[s]' (Site 2, C04), described beginning each ReSPECT conversation by framing it as common and routine, to pre-empt patients' upset reactions. Another consultant, who described ReSPECT conversations as 'emotionally very draining', conducted repeated ReSPECT conversations to manage relatives' distress:

> So if you can get some background knowledge, and if they are so in shock that they can't take anything in then it's okay to come back another time. (…) I would prefer to sit away in a, in a room together with a nurse accompanying me, so that you've got a bit of time to yourself and make sure that they know you've got time to listen to them and questions and things. (Site 1, C06).

The availability of sufficient time and adequate space influenced consultants' capacity to conduct ReSPECT conversations. Many conversations, of necessity, took place during ward rounds, and the crowdedness, urgent pace and lack of privacy in acute wards limited clinicians' ability to conduct the in-depth ReSPECT conversations they envisioned as appropriate. This consultant, for example, argued that ReSPECT conversations necessitated the quiet environment of the patient's home or GP surgery:

> …this is quite a serious and significant discussion that should not take place in a very busy, busy place. It should either happen when the patient is comfortable in their own home, or, or they have gone to see their, gone to see their GP… (Site 2, C01)

Hectic ward environments, in this consultant's experience, implicated an urgency and sensory onslaught that, together with patients' acute conditions, led to compromised conversations. The lack of sufficient time

to conduct ReSPECT conversations in acute care wards was a pervasive concern across the sample:

> …it takes time and it sort of stirs up emotions both in you and in the patient (…) so it can be very difficult, mmm, not least because you want to do it well and yet you know we were on a ward round which isn't an ideal kind of, ideally you'd come back and spend 20 minutes with each of them wouldn't you and their families and talk to them at some length. (Site 1, C09)

Notably, this consultant suggested that lack of time was not simply a logistical issue, but a factor that reduced the ability to conduct careful ReSPECT conversations and manage the emotions that arose during ReSPECT conversations.

## Theme 3: reaching ReSPECT decisions: involving versus informing

The extent to which ReSPECT conversations engaged with patients' wishes depended on consultants' clarity or uncertainty about patients' trajectories. When consultants had clear predictions for patients' short-term prognoses, they tended to lead ReSPECT conversations, taking an informative and persuasive stance. For example, when asked by the researcher, 'Are there times when you find yourself pushing the discussion in a particular way?', this consultant responded:

> Yes, I think if you genuinely feel that it would be completely futile and that you would only be prolonging an unpleasant death then yes, you do, you do tend to push the discussion in one way or another. (Site 1, C04)

Consultants often used words such as 'futile', 'frail' or 'comorbid' when describing cases in which they took a persuasive stance. Futility, as consultants framed it, foreclosed discussion of patient preferences. The conversation focused on patient preferences only when consultants were uncertain about a patient's trajectory:

> I think that the times where it's very important to discuss with a patient whether they would be appropriate for resuscitation is if it's a patient that maybe is potentially a candidate for intensive care, Level 3 care, that isn't so frail and co-morbid that we feel it would be utterly futile. (Site 2, C02)

Because they approached ReSPECT conversations according to perceptions of prognostic clarity and uncertainty, many consultants described the ReSPECT conversations in which they typically engaged—conversations with patients at imminent risk—as processes of navigation and persuasion. For example, one consultant described handling a patient's son's concerns by 'steer(ing)' the conversation:

> I went in with quite clear views of what had to be done and as you say the patient's son started to suggest that 'actually he would want to be resuscitated wouldn't

you Dad' mmm… and I gently had to steer him away to explain why I didn't think that would be a very good idea. (Site 1, C09)

As described by consultants, the need to persuade some patients and relatives was the main challenge in the ReSPECT process. To foreclose possibilities for disagreement, some consultants described structuring ReSPECT conversations to clarify which medical procedures would be undertaken:

I think a general structure is this is what's wrong, this is what we will do and this is what we won't do and if they are going to be relevant things like feeding, normal ITU, critical care I think these things need to be discussed. (Site 1, C01)

As this consultant explained, while they clarified that medical decisions were not open for discussion, they attempted to elicit patients' views during the ReSPECT conversation and integrate these into their decision-making processes:

when we are looking at what I think we can do medically we have to take into account what the patient believes (and) how they live their life… (Site 1, C01)

In line with structuring conversations to foreclose debate about medical decisions, some consultants described the ReSPECT conversation as centrally concerned with informing patients and relatives, rather than involving them in medical decision-making. For example, this consultant used the word 'disclose' to describe the function of ReSPECT conversations:

I still believe it's a medical decision and it's a good practice to inform the patient and their family. So, ultimately, the decision is mine, but I have to disclose my decision to the patient and their family. (Site 2, C06)

In other interviews, consultants suggested that, as part of the ReSPECT conversation, clinicians should state explicitly that they are informing patients and relatives about a medical decision, rather than seeking their opinion or approval. One consultant, for example, said that, when conducting a ReSPECT conversation with the relatives of a patient without capacity, one must clarify the relatives' role is to provide contextualising information and ask questions, rather than be actively involved in decision-making:

I think doctors in particular need to be clear, they're not handing over the decision making to a family member, they are still responsible for the decision but they're ensuring it's made, as far as possible, in line with what the patient would want. (Site 1, C07)

Another consultant said the ReSPECT form itself, in providing space for patient input, needed to be mediated with care, to avoid conveying that medical decisions required relatives' approval:

I will normally say that the final decision is a medical decision… 'cause the relatives say 'oh you know I need to check with my brother' when I said that 'I am informing you and just making sure you're aware that this is the reason why we are doing it'. (Site 1, C03)

Along similar lines, a consultant suggested that foregrounding patient views in the ReSPECT conversation was potentially detrimental, as it could place an undue burden on patients or lead to false hope:

If it's bleeding obvious what can and can't be offered medically then, then you have to be really careful about getting the patient to express about what they want. (…) It has the ironic effect of making them feel more ignored than they would be if, if you just gently explained what is and isn't possible. (Site 1, C09)

According to this consultant, asking patients to express their wishes unreservedly was counterproductive. Instead, this consultant argued, doctors should clarify medical possibilities and impossibilities, not place patients in the vulnerable position of having their wishes denied and their hopes deflated.

Consultants cited clear and careful communication about the finality of medical decisions as a source of comfort to patients. Describing how they would structure a ReSPECT conversation, one consultant related a hypothetical scenario in which a patient aged 82 years was diagnosed with terminal cancer. In this scenario, they said, they would relate the news to the patient, cite the evidence (as provided by blood tests) and explain what treatments will and will not be offered. Using the second person singular, the consultant described what they would say to this hypothetical patient:

Our aim will be to keep you comfortable, to support you through this. If you have any pain we will, we will control it with strong painkillers. If you have any sickness we will do that. If the time comes and if you stop breathing, or if your heart stops pumping blood… we will not be doing resuscitations, or we will not jump on your chest and perform cardiac compressions because it's not going to work. We will let you go in dignity and respect, and we will support you in that process. We will make sure your family's around you if we can. (Site 2, C03)

Reflecting on this scenario, they said this approach 'reassured' patients:

…if you're very clear to them then they can decide whether they want to be at home, whether they want to be in the hospital. And it just helps them. And if you're quite open to them, they will openly ask you questions and it just makes things easy. (Site 2, C03)

While most consultants shared a directive approach to the ReSPECT conversation, particularly in cases where they deemed resuscitation 'futile', it was not the default option for all. One consultant, for example, conceptualised

the ReSPECT conversation as 'patient-centred' and as a dialogic process towards a shared decision:

> So you start off by, by getting the patient to, to give their thoughts on what they would or wouldn't like. And that allows you to, to guide the final decision. Perhaps that's not, so it's not necessarily the patient starting with it. But you do it together. (Site 2, C05)

Framing the ReSPECT conversation as a dialogue did not preclude medical decision-making. Elsewhere in the interview, this consultant said they initiated ReSPECT conversations with patients they thought should not be for resuscitation. However, this consultant understood the ReSPECT process as complex, often comprised of multiple conversations with clinical and familial actors, building up to a shared decision. This process, they explained, led to deeper understanding and decisions that empowered patients, especially those who decided to forgo future critical care interventions.

## DISCUSSION

Our analysis found that the management of uncertainty about prognoses and patients' and relatives' emotional reactions is central to consultants' experiences of ReSPECT conversations. When determining when and with whom to conduct ReSPECT conversations, consultants rely on their predictions of a patient's short-term prognosis, prioritising patients for whom they are certain treatment escalation would not be medically indicated. When patients lack capacity, consultants also time conversations to coincide with the presence of patients' relatives, underscoring the importance of involving next of kin in these conversations, as specified in English law.[21] When determining which clinician should conduct a ReSPECT conversation and how the conversation should be framed, consultants seek to maximise rapport and minimise distress, sometimes avoiding or deferring conversations to manage uncertainty about patients' and relatives' emotional reactions. When deciding whether ReSPECT conversations should inform patients about a clinical decision or involve them in decision-making, consultants rely on their clarity or uncertainty about patients' trajectories. Thus, consultants' decisions about with whom to have RESPECT conversations, when to have these conversations and whether to frame these conversations as explaining medical decisions or as eliciting patients' preferences are driven by consultants' degrees of uncertainty about prognoses, reactions and outcomes. Throughout, the time-pressured and busy environments of acute care wards influence consultants' decisions about which conversations to prioritise and their experiences of rapport with patients.

Many of the findings are consistent with earlier studies on clinicians' experiences of barriers to DNACPR[10 22] and advance care planning (ACP) processes.[23 24] Notably, the findings resonate with a recent systematic review of qualitative studies on the implementation of ACPs, which found that clinicians' uncertainty about prognoses, clinicians' uncertainty about patients' and relatives' reactions to ACP and structural constraints related to the clinical environment all constituted barriers to ACP processes.[25] The central role of uncertainty in ReSPECT conversations both resonates with and diverges from previous research in ways that implicate features particular to ECTPs. Earlier studies have found that negotiating uncertainty is central to medical decision-making and clinical care, particularly when clinicians translate complex population-level evidence to individual prognosis and treatment.[26 27] Nonetheless, when communicating with patients, clinicians often provide reassurance through discursive modes that convey more certainty than is warranted.[28] The present analysis finds that, when conducting ReSPECT conversations, particularly with patients whose immediate trajectories are unclear, some consultants present patients and relatives with possible scenarios of future deterioration, to involve them in the decision-making process. Yet these expressions of uncertainty about prognosis and treatment, while consistent with the goals of the ECTP, sometimes clash with patients' and relatives' expectations of reassurance, clinical certainty and definitive knowledge. Previous research has suggested that clinicians can frame expressions of uncertainty productively, as an opening to shared decision-making discussions with patients.[29] Based on the present study's findings, training clinicians in how to frame uncertainty as a conversational prompt may be of particular importance in the implementation of ReSPECT.

Notably, consultants explained how they decide when, with whom and how to conduct a ReSPECT conversation through keywords which included, among others, 'frail', 'futile' and 'co-morbid'. Such keywords may serve as shorthand for clinicians' ethical stance on trajectories of treatment, although 'frail' and 'co-morbid' may also express clinical assessment. The use of such keywords without reference to clinical assessments may therefore be potentially problematic; 'futility', in particular, has been subject to debate within the medical ethics literature, with some authors arguing that the use of this term, for which no consensus definition exists, can muddle decision-making and hinder patient autonomy.[30] Previous research has found that, on DNACPR forms, clinicians entered keywords such as 'frailty' and 'futility' to justify DNACPR decisions.[9] This analysis suggests that clinicians continue to employ these keywords. How clinicians are using these keywords in the context of ReSPECT conversations warrants further exploration.

One aim of the ReSPECT process is to move discussions of future emergency treatment from a focus on CPR to broader considerations of potential treatments. Our analysis shows that some consultants are broadening these discussions. However, in the early adoption phase of ReSPECT, it seems that many conversations continue to centre on decision-making about CPR. In part, this may be related to consultants' prioritising of ReSPECT conversations with patients for whom CPR would not be

medically indicated. As the data were collected at a relatively early stage of ReSPECT implementation, it is also possible that clinicians had not yet made the conceptual shift from a DNACPR form to the more holistic approach of the ReSPECT process. Similarly, ReSPECT's key aim—to encourage a patient-centred approach to emergency care treatment planning by prompting patients' explicit involvement in the discussion—was not often realised. This was exemplified by the finding that many of the participating consultants used ReSPECT conversations to inform patients or their relatives about a clinical decision, or to steer them towards a particular decision, rather than engage them in a more open-ended discussion of their wishes and preferences. Moreover, the consultants' focus on patients for whom treatment escalation was not medically indicated also means that patients for whom treatment escalation is medically indicated, but who may wish to refuse these treatments, may not be given the opportunity to have their wishes respected. This suggests that, at early stages of implementation, the potential of ReSPECT to provide a more holistic patient-centred approach to decision-making had not yet been realised fully.

A particular strength of the analysis is the inclusion of consultants from 10 medical, orthopaedic and surgical specialties. This enables the representation of diverse secondary care environments and clinical attitudes to emergency and advance care planning. Additionally, through the study's two-stage design, whereby each consultant is first shadowed during a ward round and then interviewed, the analysis allows for an in-depth discussion of ReSPECT conversations in relation to cases observed by the researcher, thus grounding the interviews in specific and varied case examples. The analysis is limited by its focus on consultants. In both sites, consultants were responsible for signing ReSPECT forms; however, as junior doctors and nurses might take part in ReSPECT conversations, it would have been valuable to include their perspectives and experiences. Finally, as the interviews took place within the first year of ReSPECT implementation in both sites, some findings might reflect experiences related to early implementation.

## CONCLUSION
The management of uncertainty about prognoses and patients' emotional reactions is central to secondary care consultants' experiences of ReSPECT conversations. Time constraints and busy ward environments interweave with uncertainty to influence clinicians' decisions about which ReSPECT conversations to prioritise, as does the need to minimise the distress experienced by patients and their relatives and maximise rapport. While some consultants are using the ReSPECT process to broaden conversations about future emergency care treatment plans, many still focus on the decision regarding cardiopulmonary resuscitation. Additionally, conversations often focus more on communicating and explaining clinical recommendations to patients and their families rather than

exploring the patients' values and preferences to inform the decision. This suggests that the aims of the ReSPECT process are yet to be fully realised. Implementation of the ReSPECT process is still in its relatively early stages and our findings may therefore be useful to clinicians and organisations implementing ReSPECT, for example, through informing training on how to conduct ReSPECT conversations while facing uncertainty. Further research should explore how clinicians communicate uncertainty, how patients and families experience uncertainty and how clinicians' experiences of uncertainty relate to the words and values they employ when engaging in the ReSPECT process.

**Contributors** KE conducted data analysis and drafted the manuscript. A-MS, FG and CH designed the study, supervised the data collection, contributed to data analysis and commented on manuscript drafts. GDP and KC contributed to study design, assisted with accessing the field and commented on manuscript drafts. CO collected the data and commented on manuscript drafts. All authors reviewed and approved the final version of the manuscript.

**Funding** This article presents independent research funded by the National Institute for Health Research (NIHR) under the Health Services and Delivery Research programme (project number 15/15/09). KC is supported by an NIHR postdoctoral research fellowship.

**Disclaimer** The views expressed in this publication are those of the authors and not necessarily those of the NIHR or the Department of Health and Social Care.

**Competing interests** CH is a member of the ReSPECT national working group and was involved in the evaluation of ReSPECT. GDP is a member of the ReSPECT national working group and held a leading role in the development of ReSPECT; however, GDP was not involved in data collection or analysis related to the present study. A-MS, FG, CH, KC and GDP received grants from the UK National Institute of Health Research during the study.

**Patient consent for publication** Not required.

**Ethics approval** The study received ethics approval from the NRES Committee, West Midlands—Coventry and Warwickshire (REC reference: 17/WM/0134).

**Provenance and peer review** Not commissioned; externally peer reviewed.

**ORCID iDs**
Karin Eli http://orcid.org/0000-0001-9132-8404
Cynthia Ochieng http://orcid.org/0000-0002-5574-6059
Claire Hawkes http://orcid.org/0000-0001-8236-3558
Gavin D Perkins http://orcid.org/0000-0003-3027-7548
Keith Couper http://orcid.org/0000-0003-2123-2022
Frances Griffiths http://orcid.org/0000-0002-4173-1438
Anne-Marie Slowther https://orcid.org/0000-0002-3338-8457

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
