## [Reviewer comments · BMJ Open]

ARTICLE DETAILS

TITLE (PROVISIONAL)	Secondary Care Consultant Clinicians' Experiences of Conducting Emergency Care and Treatment Planning Conversations in England: An Interview-Based Analysis
AUTHORS	Eli, Karin; Ochieng, Cynthia; Hawkes, Claire; Perkins, Gavin; Couper, Keith; Griffiths, Frances; Slowther, Anne-Marie

VERSION 1 – REVIEW

REVIEWER	Professor Caroline Nicholson University of Surrey England
REVIEW RETURNED	11-Jun-2019

GENERAL COMMENTS	This is an important and timely study with very rich data . In my opinion the points below require clarification and attention to strengthen the paper : Title- should this be secondary care consultant clinicians-consultants have a particular expertise and status which I think makes their experience different from others in the MDT. Background- This is important but I wonder if it could be shortened to give more time and weight to the discussion? Methods- How were consultants identified- (Sample) how many agreed to take part (Participants) Shadowed observation requires more explanation and referencing. How were the three cases selected ? The semi-structured interview schedule - how was this designed and was there any testing of the schedule prior to the study?- It would be helpful to put in an appendix PPI- Did the PPI group input into the design or conduct of the study at all e.g. the ethics of somebody else observing this sensitive conversation ? Ethics- This requires more work - Was the patient involved at all? How did you inform the ward that shadowed observation was happening ? Results- These are important themes however there is overlap between theme one and two and the involvement of family members- The use of the word clinician is confusing- these are all consultants with a particular seniority, power and expertise-
--

	Discussion: It is not always clear what you are saying "throughout, the spatial and temporal constraints of the acute care ward influence clinicians' decisions about which conversations to prioritize and their experiences of rapport with patients" - Anthropological theory can be very useful but needs to be more fully explained IS this about the experience of consultant clinicians or how consultant clinicians implement the Respect document (are they different?) Lund S, Richardson A, May C (2015) Barriers to Advance Care Planning at the End of Life: An Explanatory Systematic Review of Implementation Studies. PLoS ONE 10(2): e0116629. https://doi.org/10.1371/journal.pone.0116629 is I think a useful paper for you to look at- If this is about managing clinical uncertainty as your conclusion suggests you may wish to look how others have discussed medical uncertainty and its implications for practice (eg Wray 2015 doi/pdf/10.4300/JGME-D-14-00638.1) The first statement is a repetition of the background- Given the importance of family and who to involve in the process this is minimal in the discussion.
--	--

REVIEWER	Jürgen in der Schmitt Institute of General Practice Medical Faculty, Heinrich-Heine-University of Düsseldorf Germany
REVIEW RETURNED	08-Jul-2019

GENERAL COMMENTS	Secondary Care Clinicians' Experiences of Conducting Emergency and Advance Care Planning Conversations: An Interview-Based Analysis Karin Eli et al. MS-ID: bmjopen-2019-031633 Eli and colleagues have followed 15 medical and surgical consultants on their ward rounds where the latter often led ReSPECT conversations, i.e. conversations using the ReSPECT tool that encourages clarification of resuscitation status during a hospital stay. After the wardrounds and thus inspired by the real patient encounters, the consultants explained in interviews their views on and experiences with ReSPECT conversations. As far as I can judge, this qualitative interview study is well done methodologically, the results are transparently presented, and they allow a thorough insight into the interviewed consultants views and attitudes. As detailed below, I am not happy with the interpretation of the results in the discussion section. In my eyes, it is both justified and indeed important that this study, after major review of the background and discussion sections, is being published and discussed. Background: I am not happy with the first sentences of the background, see line-by-line critique below. The confusion that I describe below is of significance for the discussion section since it allows the consideration whether the authors themselves
---

sufficiently distinguish between what is medically indicated on the one hand, and what is wanted or preferred by the patient on the other hand.

Also in the background, the authors describe the tool ReSPECT. Please see my comment in the below line-per-line critique. Again, this is relevant for the pivotal unclarity of this study, i.e. the relation between medical indication (what can still be justified because of an at least minute chance of success) and patient preference (what of the medically justifiable is wanted by the patient). The authors praise their instrument as „emphasising patient involvement“, and „recording patients' wishes“. Ironically, in their results they report that exactly this does not happen because (most) physicians see the tool as documenting what they view as mainly medical decisions that patients and family, if anything, should be informed about. I would invite the authors to reflect in the first place whether their instrument, the ReSPECT tool, does in itself truly encourage patient involvement. I would say it doesn't, since the decisions on emergency care and treatment are left at the discretion of the physician („clinical recommendations“) where they may or may not consider the „personal preferences to guide this plan“.

Methods: I confess that I am not deeply involved in qualitative methods in general, and in thematic analysis in particular. From what I can see, also considering the SRQR guidelines, the method for data collection and analysis is chosen appropriately, thoroughly described, and practiced in a transparent, consistent way.

Results: I gladly confess that I am very impressed with the results section. The authors have beautifully structured their results (by three themes and a number of sub-themes each), and they perfectly illustrate each sub-theme by fitting examples from the empiric material. They thus allow the reader to appreciate their findings on his or her own.

Discussion: In my view, the reported results regarding theme 3 are a sensation, just phenomenal. They show that most of the involved consultants have not fully understood, appreciated or embraced the concept of shared decision making, a process aimed at enabling individuals to make the treatment decisions that suit them best. There is a flagrant contrast between the above quoted claim of „patient involvement“ (even if I hold that already the ReSPECT concept, at close look, does not truly substantiate this claim), and the reported view of many doctors that they are mainly informing patients and relatives about „medical decisions“.

In this perspective, the reported results on the theme „uncertainty management and catalyst for the decision“ fit perfectly into the pattern. Obviously, most interviewed consultants approach patients only when they „predict a trajectory of deterioration.“ I would expect from the discussion to make transparent that this is a code (similar to the codes „frail“ or „co-morbid“ commented in the discussion), essentially implying that ReSPECT conversations are mainly offered to patients in whom at least CPR is no longer medically indicated, and certainly in the view of the clinicians also other intensive care measures should rather be withheld.

I miss in the discussion a prominent clarification that the views expressed by the clinicians may collide with the principle of

autonomy in general, and that they certainly represent a misunderstanding of Advance Care Planning (ACP) in particular. ACP has been consistently defined in countless publications over many years since the 1990ies, recently (2017) in two consensus reports (by Rietjens et al, and Sudore et al), and it always consistently implies that it is the patient who makes the decision – as long as a treatment attempt is medically justified, i.e. as long as there is not certainty that the patients' treatment goal cannot be reached with an attempt of this treatment. So I would strongly advise that the term ACP is not used in the entire paper for a process that at least according to the authors' results mostly (I see the few exceptions) burns down to informing (pre-) terminal patients about the medical necessity to forgo certain life-sustaining options.

Now I realise that there is futile treatment, and being a palliative care physician myself it is my least intention to encourage patients to insist on treatments that are no longer indicated. The concept of futility, however, is extremely complex and has been debated in depth in the past – a debate that I miss a reference to in the first background paragraph, and that I also miss to be deeply appreciated in the discussion. It is altogether not clear what the consultants reported to view the ReSPECT documentation as „medical decisions“ understand to be the precise prognosis in these cases, and in how far they appreciate this complex discussion, when they speak of a treatment as „futile“. It is not a question that the success rates of CPR, for example, can become very low, certainly < 5%, if certain comorbidities exist, and there is sound empirical data to predict this. But what are the data to support a decision not to take a patient on an ICU again, or to withhold intubation and ventilation? No question that frailty and comorbidities do substantially lower the success rate of intensive care attempts. But what if the patient does not share the consultant's value judgment that a certain low probability of success must be viewed as futile (or too low, as it were)? In my eyes, these are questions that a discussion of these exciting results should pick up.

Concurrently, the discussion should analyse what it factually means for the ReSPECT tool in the end if, as shown by the authors in the results section, (most) physicians use it (most of the time) to approach patients when their prognosis is dim, and thus intensive life-sustaining treatment is of little prospect if not futile. This may mean that the instrument is not at all understood, and used, to encourage patient involvement, but indeed just to „inform about medical decisions“.

Not only some severely ill patients may thus become subject of neo-paternalistic physician judgments; also many patients whom physicians view to be too good to forgo life-sustaining treatment may rarely be approached and offered ReSPECT conversations (using typical rationalising justifications such as limited time resources, or patient misperceptions of such conversations offers), and thus never receive the opportunity to reject potentially life-sustaining treatments that the interviewed consultants may deem appropriate without asking, while the patients might have a different view if they were enabled to make an informed decision in the true sense of Advance Care Planning. The authors can easily check whether my conjectures are substantiated: If I am right, most ReSPECT forms filled out by the consultants of this study

should conclude that CPR be withheld, and that focus should be on symptom control. Such a (mis)understanding of the tool deserves in my eyes, however, an appreciation and analysis in the discussion.

Below are examples for sentences in the discussion that could, in my opinion, be easily omitted, thus making room for a more prominent appreciation of the patient involvement & autonomy versus physician paternalism issue set out above.

Line-by-line Critique

P. 3, L 4ff.

The first two sentences are redundant in the sense that their content has a large intersection, and they confuse rather than clarify this important subject. The first sentence is in itself incorrect or at least misleading: DNACPR orders allow practitioners to withhold CPR when the prognosis is futile OR when the patient refuses CPR (independent of prognosis). The concept of futility is complex and demanding, and its debate has a number of important implications. This debate should be referred to in this introduction, rather than using the unprecise phrase „when cardiac arrest occurs as part of the person’s natural and irreversible dying process“. The latter wording suggests that this point can be easily determined, or is a matter of scientific determination; as the futility debate has taught us, at closer sight this is often not the case. But also the second sentence is not a sufficient clarification: I cannot recognise a convincing third category between medical futility on the one hand, and patients’ autonomous decision on the other. The additional case suggested by the authors, „when the burdens ... outweigh ... benefits“, may not be seen as an alternative case versus „when a patient requests not to be resuscitated“, but rather the patient’s rationale in this latter case. A clear, precise introduction in this subject, however, is highly desirable in order not to perpetuate and deepen the ambiguity and shortcomings that the authors appropriately report in the remaining paragraph.

P3, L 28-56

ReSPECT is a remarkable project, and certainly deserves great appreciation. An obvious strength is that the ECTP is not simply made available to patients, but that it is closely linked with a communication process.

I would, however, invite the authors to consider a comparison with the US POLST program (in combination with ACP as practiced in Respecting Choices), or also the German emergency form that is also part of a larger ACP communication process (available in German only: Nauck et al. in AINS 2018). The difference is that while the ReSPECT ECTP is a non-binding treatment recommendation by a clinician, based on a conversation with a patient, the POLST and even more so the German „No represent an autonomous and detailed decision by the patient, enabled by a physician ensuring a sound process of an informed refusal. In other words, while it may be right that ReSPECT „emphasizes patient involvement“, it possibly does not so much „record patient’s wishes“ but rather „physicians’ recommendations“, based on their understanding of patient’s values and preferences. In order to truly elicit and document patients’ wishes on CPR and other specific

life-sustaining measures, clinicians would have to understand their role in supporting patients' decisions, informing them about the empirical chances and risks of CPR attempts, and discussing what these mean to this patient. Perhaps the authors want to consider these deliberations in their description of ReSPECT.

P 4, L 40ff

Thank you for the thorough methods' description. I find it hard to appreciate why the authors „decided to focus the analysis on the three overarching themes concerned with the ReSPECT conversation“. To me it seems that including the two other themes, focusing on clinicians' value judgments, and on the ReSPECT form resp, in this analysis would have been helpful or even important, especially the first since clinicians' value judgments may shape or explain much of their conversation. Perhaps the authors can include their rationale for this decision.

P8, L 36

Maybe I am mistaken, but should it not be: „... I am not looking at her AS HER responsible consultant?“

P12, L 7

What are „conversational templates“ in this context? If the authors don't think that my language barrier is the problem here, they may want to consider using a different phrase.

P13, L 54

Is it really about clinicians' experiences of the ReSPECT process, or not rather about clinicians perspectives and views on the process? Surely the experiences contribute to views to varying extends, but what I have mainly read are views or attitudes.

P13, L 55-58

„the findings show that ReSPECT conversations relate to overall treatment plans and are not limited to resuscitation decisions“: I cannot see that this alleged finding, put on No 1 in the discussion's first paragraph, is supported by any of the explicitly reported results, certainly not the subthemes. I agree that some of the quotations are compatible with this interpretation, but I find it difficult to summarise a result at this exposed spot (= first paragraph of discussion) that may (or may not) only be concluded between the lines.

P15, L 22-27

„Because the ReSPECT process requires clinicians to hold conversations with patients or their relatives before recording a decision about emergency care and treatments, it may encourage clinicians to articulate decision-making processes and to develop strategies for managing and communicating uncertainty“: This sentence is not derived from the results and thus is not an apt element of a conclusion; rather, it seems an appraisal of the tool that according to the COI-Statement, 2 of the authors have developed. The following sentence, beginning with „These findings“, has no clear reference, since the quoted sentence does not refer to a finding, but praises the ReSPECT tool on general grounds. In contrast, while the ReSPECT form does include a section where the physician is expected to „provide clinical guidance on specific interventions that may or may not be wanted or clinically appropriate, including being taken or admitted to

	hospital +/- receiving life support“, I haven’t found a single report in the study about a concrete statement referring to any other medical treatment option than CPR.
--	---

VERSION 1 – AUTHOR RESPONSE

Reviewer: 1

1. This is an important and timely study with very rich data. In my opinion the points below require clarification and attention to strengthen the paper:

Response: Thank you for your encouraging feedback. We have revised the manuscript in accordance with your suggestions, as detailed below.

2. Title - should this be secondary care consultant clinicians - consultants have a particular expertise and status which I think makes their experience different from others in the MDT.

Response: We have changed the title to “Secondary Care Consultant Clinicians’ Experiences of Conducting Emergency Care and Treatment Planning Conversations in England: An Interview-Based Analysis”

3. Background- This is important but I wonder if it could be shortened to give more time and weight to the discussion?

Response: We have edited the first three paragraphs of the introduction to reduce their length.

Methods-

4. How were consultants identified- (Sample) how many agreed to take part (Participants)?

Response: We have clarified this in the methods section:

“Potential participants were identified by the local principal investigator at each of the participating hospitals through purposive sampling designed to represent a range of views about the ReSPECT process, as well as a diversity of clinical areas that could be replicated across NHS trusts (three medical specialities, a surgical speciality, and orthopaedics). The local PIs or research nurses asked for volunteer participants from these specialities and the study’s research fellow scheduled ward round observations directly with the participating consultants, to ensure that observations did not place an undue burden on their clinical practice” (p. 4, lines 98-104).

5. Shadowed observation requires more explanation and referencing. How were the three cases selected?

Response: We have provided more explanation of shadowing, citing Czarniawska-Joerges' (2007) book on shadowing as a qualitative research method, and Davies' (2003) and Rice's (2010) studies, in which they used shadowing to explore doctors' experiences. We have also added a description of how cases were selected:

“if the researcher observed three ReSPECT conversations (which was the maximum she had observed in any of her observation sessions), she selected these three cases for discussion during the interviews. If she observed fewer than three ReSPECT conversations, she selected one or more cases where she thought a ReSPECT conversation might have been appropriate, to explore with the clinician why they chose not to hold a ReSPECT conversation in those cases” (p. 4, lines 119-124).

6. The semi-structured interview schedule - how was this designed and was there any testing of the schedule prior to the study?- It would be helpful to put in an appendix

Response: We have clarified that:

“The interview topic areas were developed based on the study's research questions and the literature, and the observation and interview approach was checked with members of the study team with relevant clinical experience” (p. 4, lines 124-126).

PPI-

7. Did the PPI group input into the design or conduct of the study at all e.g. the ethics of somebody else observing this sensitive conversation?

Response: The PPI group helped with the grant application and discussed the consenting models for the study. We have added these details to the section on PPI.

Ethics-

8. This requires more work - Was the patient involved at all? How did you inform the ward that shadowed observation was happening ?

Response: We now clarify in the methods section that:

“To ensure that patients, relatives, and staff were aware that observations were taking place, study posters were displayed in the selected wards, and the research fellow wore a scrubs uniform top with the word ‘researcher’ printed clearly on both the front and the back. During each shadowing period the participating consultant introduced the researcher to each patient (and family if appropriate) and informed

them that they could request that the researcher leave if they wished. A brief information leaflet was left with the patient” (p. 4, lines 110-116).

Results-

9. These are important themes however there is overlap between theme one and two and the involvement of family members

Response: We have edited the text to clarify that whereas theme one refers to the involvement of family members as related to timing, theme two relates to emotions.

10. The use of the word clinician is confusing- these are all consultants with a particular seniority, power and expertise

Response: We have replaced the word “clinicians” with “consultants” throughout the manuscript, when referring to the participants.

Discussion:

11. It is not always clear what you are saying "throughout, the spatial and temporal constraints of the acute care ward influence clinicians' decisions about which conversations to prioritize and their experiences of rapport with patients" - Anthropological theory can be very useful but needs to be more fully explained

Response: We have edited this to remove anthropological jargon. The sentence now reads:
“Throughout, the time-pressured and busy environments of acute care wards influence consultants' decisions about which conversations to prioritize and their experiences of rapport with patients” (p. 14, lines 568-570).

12. IS this about the experience of consultant clinicians or how consultant clinicians implement the Respect document (are they different?)

Response: We have removed the statement at the top of the discussion that refers to how consultants implement ReSPECT, in order to maintain the manuscript's focus on the experiences of consultants.

13. Lund S, Richardson A, May C (2015) Barriers to Advance Care Planning at the End of Life: An Explanatory Systematic Review of Implementation Studies. PLoS ONE 10(2): e0116629. <https://doi.org/10.1371/journal.pone.0116629>

is I think a useful paper for you to look at-

If this is about managing clinical uncertainty as your conclusion suggests you may wish to look how others have discussed medical uncertainty and its implications for practice (eg Wray 2015 doi/pdf/10.4300/JGME-D-14-00638.1)

Response: Thank you for recommending these helpful references. We have added them both to enhance our discussion.

14. The first statement is a repetition of the background - Given the importance of family and who to involve in the process this is minimal in the discussion.

Response: We have deleted the first sentence of the discussion. We have also added the following statement to the first paragraph of the discussion, to emphasize further the importance of family involvement:

“Moreover, where patients without capacity are concerned, consultants time conversations to coincide with the presence of patients’ relatives, underscoring the importance of involving an individual close to the patient in these conversations, as specified in English law” (p. 14, lines 557-559).

Reviewer 2

1. Eli and colleagues have followed 15 medical and surgical consultants on their ward rounds where the latter often led ReSPECT conversations, i.e. conversations using the ReSPECT tool that encourages clarification of resuscitation status during a hospital stay. After the wardrounds and thus inspired by the real patient encounters, the consultants explained in interviews their views on and experiences with ReSPECT conversations. As far as I can judge, this qualitative interview study is well done methodologically, the results are transparently presented, and they allow a thorough insight into the interviewed consultants views and attitudes. As detailed below, I am not happy with the interpretation of the results in the discussion section. In my eyes, it is both justified and indeed important that this study, after major review of the background and discussion sections, is being published and discussed.

Response: Thank you for your thorough and very helpful review. We have addressed your comments in detail, both in the manuscript and in the responses we follow. We hope you will find our manuscript has been strengthened as a result.

2. **Background:** I am not happy with the first sentences of the background, see line-by-line critique below. The confusion that I describe below is of significance for the discussion section since it allows the consideration whether the authors themselves sufficiently distinguish

between what is medically indicated on the one hand, and what is wanted or preferred by the patient on the other hand. Also in the background, the authors describe the tool ReSPECT. Please see my comment in the below line-perline critique. Again, this is relevant for the pivotal unclarity of this study, i.e. the relation between medical indication (what can still be justified because of an at least minute chance of success) and patient preference (what of the medically justifiable is wanted by the patient).

Response: We agree that the first sentence of the background section is problematic; we have now removed it. Please see our responses to the next comment, and to the line-by-line critique, where we address the relationship between medical indication and patient preference in greater detail.

3. The authors praise their instrument as „emphasising patient involvement“, and „recording patients‘ wishes“. Ironically, in their results they report that exactly this does not happen because (most) physicians see the tool as documenting what they view as mainly medical decisions that patients and family, if anything, should be informed about. I would invite the authors to reflect in the first place whether their instrument, the ReSPECT tool, does in itself truly encourage patient involvement. I would say it doesn't, since the decisions on emergency care and treatment are left at the discretion of the physician („clinical recommendations“) where they may or may not consider the „personal preferences to guide this plan“

Response: Thank you for this important comment. As explained in paragraph two of our paper, ReSPECT is not our instrument. We have edited the final paragraph of the introduction to clarify the nature and purpose of our study:

“we report findings from interviews with secondary-care consultant clinicians in two NHS organisations that had recently implemented ReSPECT, exploring why, when, and with whom they choose to have ReSPECT conversations. Our aim is to inform future development of the process and the current implementation across the NHS and to provide focus to further qualitative research on how ReSPECT becomes integrated into health professionals' practice” (p. 3, lines 84-88).

To increase clarity about our team's involvement with ReSPECT, we have added the following statement to the competing interests section:

“CH is a member of the ReSPECT national working group and was involved in the evaluation of ReSPECT. GDP is a member of the ReSPECT national working group and held a leading role in the development of ReSPECT; however, GDP was not involved in data collection or analysis related to the present study” (p. 19, lines 734-738).

We acknowledge that ReSPECT is designed to inform clinical recommendations through patient involvement, rather than privilege patients' choices. One of our findings is that while ReSPECT does prompt consultants to involve patients and/or their relatives in conversations about CPR and other emergency treatments, the conversations are often aimed at informing patients and/or relatives about medical decisions, rather than fully involving them in decision-making.

We have amended our background section to clarify the position of ReSPECT with regard to patient involvement:

“ReSPECT is an emergency care treatment plan (ECTP) developed in response to the gaps observed in the DNACPR process. ReSPECT builds on research conducted in the US, the UK, and Canada, which found that programmes that integrate DNACPR with discussions about wider goals of treatment increase clarity about trajectories of care and reduce harm to patients. As an ECTP which records clinical recommendations that take into account patients' values and preferences, ReSPECT places resuscitation within a wider context of treatments that should or should not be considered in an emergency situation” (p. 3, lines 65-71).

ReSPECT is conceptualised as part of good clinical practice in line with professional guidance and UK law. In the UK, clinicians responsible for the care of patients who lack capacity have the legal responsibility for making decisions about treatment in accordance with the Mental Capacity Act, and based on their assessment of what is in the patient's best interests. This is often the case in an emergency situation and always the case in a cardiac arrest. Any record of an advance **refusal** of treatment must be respected. A doctor must also seek to ascertain what the patient's past and present views, wishes and values about treatment may be, in order to inform their assessment of what is in the patient's best interests. Advance care plans and patients' families can provide information to guide the doctor in these situations.

The purpose of the ReSPECT process is to enable the patient's values and preferences about the kind of care they wish to have (or not to have) in an emergency situation to be documented and to facilitate a discussion with the doctor to hopefully come to an agreed decision on what future treatments should be recommended or not. A patient's wish not to have specific treatments must be respected, but a patient cannot require a clinician to provide specific treatment if the clinician thinks it is not clinically indicated or in the patient's best interests. ReSPECT is designed to record recommendations based on clinical judgement and informed by the patient's wishes and preferences (or directed by the patient's wishes to refuse specific treatments). We have edited the second and third paragraph of the introduction and the first paragraph of the discussion to clarify this for readers:

Introduction: “As an ECTP which records clinical recommendations that take into account patients’ values and preferences, ReSPECT places resuscitation within a wider context of treatments that should or should not be considered in an emergency situation. The authors of ReSPECT emphasise that it is a process designed to guide clinicians in discussing with patients what might be optimal treatment choices for them with the ReSPECT form acting as a prompt and summary record of the discussion and its outcomes” (p. 3, lines 69-75).

Discussion: “When patients lack capacity, consultants time conversations to coincide with the presence of patients’ relatives, underscoring the importance of involving next of kin in these conversations, as specified in English law” (p. 14, lines 557-559).

4. **Methods**: I confess that I am not deeply involved in qualitative methods in general, and in thematic analysis in particular. From what I can see, also considering the SRQR guidelines, the method for data collection and analysis is chosen appropriately, thoroughly described, and practiced in a transparent, consistent way.

Response: Thank you for your positive assessment of our methods.

5. **Results**: I gladly confess that I am very impressed with the results section. The authors have beautifully structured their results (by three themes and a number of sub-themes each), and they perfectly illustrate each sub-theme by fitting examples from the empiric material. They thus allow the reader to appreciate their findings on his or her own.

Response: We are grateful for your appreciative review of our results section.

6. **Discussion**: In my view, the reported results regarding theme 3 are a sensation, just phenomenal. They show that most of the involved consultants have not fully understood, appreciated or embraced the concept of shared decision making, a process aimed at enabling individuals to make the treatment decisions that suit them best. There is a flagrant contrast between the above quoted claim of „patient involvement“ (even if I hold that already the ReSPECT concept, at close look, does not truly substantiate this claim), and the reported view of many doctors that they are mainly informing patients and relatives about „medical decisions“. In this perspective, the reported results on the theme „uncertainty management and catalyst for the decision“ fit perfectly into the pattern. Obviously, most interviewed consultants approach patients only when they „predict a trajectory of deterioration.“ I would expect from the discussion to make transparent that this is a code (similar to the codes „frail“ or „co-morbid“ commented in the discussion), essentially implying that ReSPECT conversations are mainly offered to patients in whom at least CPR is no longer medically indicated, and certainly in the view of the clinicians also other intensive care measures should rather be withheld.

Response: Thank you for highlighting this important point. We agree that our data show that consultants appear to be using the ReSPECT process to communicate medical prognosis and medical decisions rather than fully involving patients or their families in the discussion about treatment decisions. We think it is important to note that the data were collected at a relatively early stage of the ReSPECT process implementation in each hospital and that many doctors would not have made the conceptual shift from a DNACPR form to the more holistic approach of the ReSPECT process with its aim of patient involvement. Our evaluation study has continued and extends to several other hospitals and it will be interesting to see if patient involvement in the process increases as it becomes embedded in clinical practice.

Your observation that ReSPECT conversations are mainly offered to patients for whom at least CPR is not medically indicated and for whom clinicians feel that other intensive treatments should be withheld is also important. In these cases doctors may find it more difficult to involve patients in a discussion about treatment preferences when many treatments may not be medically indicated. To address this, we have added the following clause to the discussion:

“When determining when and with whom to conduct ReSPECT conversations, consultants rely on their predictions of a patient’s short-term prognosis, **prioritising patients for whom treatment escalation would not be medically indicated**” (p. 14, lines 554-557).

7. I miss in the discussion a prominent clarification that the views expressed by the clinicians may collide with the principle of autonomy in general, and that they certainly represent a misunderstanding of Advance Care Planning (ACP) in particular. ACP has been consistently defined in countless publications over many years since the 1990ies, recently (2017) in two consensus reports (by Rietjens et al, and Sudore et al), and it always consistently implies that it is the patient who makes the decision – as long as a treatment attempt is medically justified, i.e. as long as there is not certainty that the patients’ treatment goal cannot be reached with an attempt of this treatment. So I would strongly advise that the term ACP is not used in the entire paper for a process that at least according to the authors’ results mostly (I see the few exceptions) burns down to informing (pre-) terminal patients about the medical necessity to forgo certain life-sustaining options.

Response: Thank you for this important comment; we apologise for the confusion, and have now have removed “advance care planning” from the title. We now refer to ReSPECT exclusively as an ECTP throughout the manuscript. The only instances where ACP is referenced elsewhere in the manuscript appear when we compare RESPECT to ACP programs.

8. Now I realise that there is futile treatment, and being a palliative care physician myself it is my least intention to encourage patients to insist on treatments that are no longer indicated. The concept of futility, however, is extremely complex and has been debated in depth in the past – a debate that I miss a reference to in the first background paragraph, and that I also miss to be deeply appreciated in the discussion. It is altogether not clear what the consultants reported to view the ReSPECT documentation as „medical decisions“ understand to be the precise prognosis in these cases, and in how far they appreciate this complex discussion, when they speak of a treatment as „futile“. It is not a question that the success rates of CPR, for example, can become very low, certainly < 5%, if certain comorbidities exist, and there is sound empirical data to predict this. But what are the data to support a decision not to take a patient on an ICU again, or to withhold intubation and ventilation? No question that frailty and comorbidities do substantially lower the success rate of intensive care attempts. But what if the patient does not share the consultant’s value judgment that a certain low probability of success must be viewed as futile (or too low, as it were)? In my eyes, these are questions that a discussion of these exciting results should pick up.

Response: Thank for highlighting the need to consider futility more deeply. First, it is important to note that although we include several quotes in which consultants explained they did not recommend escalation of care because it would have been ‘futile’, we do not wish to imply that ‘futility’ is an acceptable shorthand in decision-making. To clarify this, we have deleted the first sentence of the introduction. In the discussion, we have added a statement on the critical consideration of futility, citing Ardagh (2000):

“Notably, consultants explained how they decide when, with whom, and how to conduct a ReSPECT conversation through keywords which include, among others, ‘frail’, ‘futile’, and ‘co-morbid’. Such keywords may serve as shorthand for clinicians’ ethical stance on trajectories of treatment, although ‘frail’ and ‘co-morbid’ may also express a clinical judgement based on assessment tools. The use of such keywords without reference to clinical assessments may therefore be potentially problematic; ‘futility’, in particular, has been subject to debate within the medical ethics literature, with some authors arguing that the use of this term, for which no consensus definition exists, can muddle decision-making and hinder patient autonomy” (p. 15, lines 591-598).

The question of how decisions are made (either in advance or in real time) about whether to admit a patient to intensive care, or commence ventilation or other intensive therapies, and the values that influence these decisions (patients, families and clinicians) is important. There is an extensive international literature on empirical studies looking at factors that influence clinician’s decisions around admission to intensive care which suggest that such decisions are influenced by clinicians’ assessment of a range of patient characteristics rather than patient preferences. Even when a patient’s preferences are taken into account it is likely that

the clinician's values will frame the presentation of evidence and potential outcomes in discussions with the patient. However, it is not possible to fully explore this complex phenomenon within the context of this paper. This issue is the subject of a separate research project undertaken by members of this research team.

9. Concurrently, the discussion should analyse what it factually means for the ReSPECT tool in the end if, as shown by the authors in the results section, (most) physicians use it (most of the time) to approach patients when their prognosis is dim, and thus intensive life-sustaining treatment is of little prospect if not futile. This may mean that the instrument is not at all understood, and used, to encourage patient involvement, but indeed just to „inform about medical decisions“. Not only some severely ill patients may thus become subject of neo-paternalistic physician judgments; also many patients whom physicians view to be too good to forgo life-sustaining treatment may rarely be approached and offered ReSPECT conversations (using typical rationalising justifications such as limited time resources, or patient misperceptions of such conversations offers), and thus never receive the opportunity to reject potentially life-sustaining treatments that the interviewed consultants may deem appropriate without asking, while the patients might have a different view if they were enabled to make an informed decision in the true sense of Advance Care Planning. The authors can easily check whether my conjectures are substantiated: If I am right, most ReSPECT forms filled out by the consultants of this study should conclude that CPR be withheld, and that focus should be on symptom control. Such a (mis)understanding of the tool deserves in my eyes, however, an appreciation and analysis in the discussion.

Response: In the third paragraph of the discussion, we have now added a reflection on the link between participating consultants' use of keywords like 'futility' and their framing of the ReSPECT conversation:

“Notably, consultants explained how they decide when, with whom, and how to conduct a ReSPECT conversation through keywords which include, among others, 'frail', 'futile', and 'co-morbid'. Such keywords may serve as shorthand for clinicians' ethical stance on trajectories of treatment, although 'frail' and 'co-morbid' may also express a clinical judgement based on assessment tools. The use of such keywords without reference to clinical assessments may therefore be potentially problematic; 'futility', in particular, has been subject to debate within the medical ethics literature, with some authors arguing that the use of this term, for which no consensus definition exists, can muddle decision-making and hinder patient autonomy” (p. 15, lines 591-598).

10. Below are examples for sentences in the discussion that could, in my opinion, be easily omitted, thus making room for a more prominent appreciation of the patient involvement & autonomy versus physician paternalism issue set out above.

Response: Thank you for these detailed comments, which we address below.

Line-by-line Critique

11. **P. 3, L 4ff.** The first two sentences are redundant in the sense that their content has a large intersection, and they confuse rather than clarify this important subject. The first sentence is in itself incorrect or at least misleading: DNACPR orders allow practitioners to withhold CPR when the prognosis is futile OR when the patient refuses CPR (independent of prognosis). The concept of futility is complex and demanding, and its debate has a number of important implications. This debate should be referred to in this introduction, rather than using the unprecise phrase „when cardiac arrest occurs as part of the person’s natural and irreversible dying process“. The latter wording suggests that this point can be easily determined, or is a matter of scientific determination; as the futility debate has taught us, at closer sight this is often not the case. But also the second sentence is not a sufficient clarification: I cannot recognise a convincing third category between medical futility on the one hand, and patients’ autonomous decision on the other. The additional case suggested by the authors, „when the burdens ...outweigh ... benefits“, may not be seen as an alternative case versus „when a patient requests not to be resuscitated“, but rather the patient’s rationale in this latter case. A clear, precise introduction in this subject, however, is highly desirable in order not to perpetuate and deepen the ambiguity and shortcomings that the authors appropriately report in the remaining paragraph.

Response: We have now deleted the first sentence of the introduction, and have modified the second sentence to clarify it:

“Most DNACPR decisions are made when clinicians determine that the burdens that cardiopulmonary resuscitation (CPR) would pose to the patient outweigh potential benefits. However, DNACPR decisions can also be made when clinicians predict that CPR would not succeed, or when a patient requests not to be resuscitated” (p. 3, lines 50-53).

12. **P3, L 28-56** ReSPECT is a remarkable project, and certainly deserves great appreciation. An obvious strength is that the ECTP is not simply made available to patients, but that it is closely linked with a communication process. I would, however, invite the authors to consider a comparison with the US POLST program (in combination with ACP as practiced in Respecting Choices), or also the German emergency form that is also part of a larger ACP communication process (available in German only: Nauck et al. in AINS 2018). The difference is that while the ReSPECT ECTP is a non-binding treatment recommendation by a clinician, based on a conversation with a patient, the POLST and even more so the German ÄNo represent an autonomous and detailed decision by the patient, enabled by a physician

ensuring a sound process of an informed refusal. In other words, while it may be right that ReSPECT „emphasizes patient involvement“, it possibly does not so much „record patient’s wishes“ but rather „physicians’ recommendations“, based on their understanding of patient’s values and preferences. In order to truly elicit and document patients’ wishes on CPR and other specific life-sustaining measures, clinicians would have to understand their role in supporting patients’ decisions, informing them about the empirical chances and risks of CPR attempts, and discussing what these mean to this patient. Perhaps the authors want to consider these deliberations in their description of ReSPECT.

Response: We now clarify in the introduction that, as an ECTP, ReSPECT is not the same as an ACP:

“As an ECTP which records clinical recommendations that take into account patients’ values and preferences, ReSPECT places resuscitation within a wider context of treatments that should or should not be considered in an emergency situation. The authors of ReSPECT emphasise that it is a process designed to guide clinicians in discussing with patients what might be optimal treatment choices for them with the ReSPECT form acting as a prompt and summary record of the discussion and its outcomes” (p. 3, lines 69-75).

13. **P 4, L 40ff** Thank you for the thorough methods’ description. I find it hard to appreciate why the authors „decided to focus the analysis on the three overarching themes concerned with the ReSPECT conversation“. To me it seems that including the two other themes, focusing on clinicians’ value judgments, and on the ReSPECT form resp, in this analysis would have been helpful or even important, especially the first since clinicians’ value judgments may shape or explain much of their conversation. Perhaps the authors can include their rationale for this decision.

Response: We now clarify in the introduction and methods section that our main interest in this analysis was to understand when, why and with whom consultants choose to hold ReSPECT conversations. As such, while value judgments are important to ReSPECT conversations (and indeed, our analysis is also informed by considering how participants use keywords such as ‘futility’, for example), we decided to keep the thematic focus on the conversation itself, as these three themes most closely responded to our main interest.

14. **P8, L 36** Maybe I am mistaken, but should it not be: „... I am not looking at her AS HER responsible consultant?“

Response: The quote as it originally appeared in our manuscript was correct. However, to reduce ambiguity, we have now added [as her] in brackets.

15. **P12, L 7** What are „conversational templates“ in this context? If the authors don't think that my language barrier is the problem here, they may want to consider using a different phrase.

Response: We have changed the wording of this clause, so that it now reads: “In line with structuring conversations to foreclose debate about medical decisions” (p. 12, line 474).

16. **P13, L 54** Is it really about clinicians' experiences of the ReSPECT process, or not rather about clinicians perspectives and views on the process? Surely the experiences contribute to views to varying extends, but what I have mainly read are views or attitudes.

Response: We have changed the first sentence of the discussion, so that it better contextualises what we mean by 'experiences':

“Our analysis found that the management of uncertainty about prognoses and patients' and relatives' emotional reactions is central to consultants' experiences of ReSPECT conversations” (p. 14, lines 553-554).

17. **P13, L 55-58** „the findings show that ReSPECT conversations relate to overall treatment plans and are not limited to resuscitation decisions“: I cannot see that this alleged finding, put on No 1 in the discussion's first paragraph, is supported by any of the explicitly reported results, certainly not the subthemes. I agree that some of the quotations are compatible with this interpretation, but I find it difficult to summarise a result at this exposed spot (= first paragraph of discussion) that may (or may not) only be concluded between the lines.

Response: Thank you for calling our attention to this. This statement was included to highlight that most clinicians spoke not only about CPR, but also about other treatment decisions (e.g., escalation to intensive care), in the context of ReSPECT, as evidenced in some of the quotes that appear in the results section. We have now removed this statement from the first paragraph of the discussion. Instead, we have added a new paragraph that discusses this finding with greater nuance and contextualizes it within the broader findings of this analysis:

“One aim of the ReSPECT process is to move discussions of future emergency treatment from a focus on CPR to broader considerations of potential treatments. Our analysis shows that some consultants are broadening these discussions. However, in the early adoption phase of ReSPECT, it seems that many conversations continue to centre on decision-making about CPR. In part, this may be related to consultants' prioritising of ReSPECT conversations with patients for whom CPR would not be medically indicated. As the data were collected at a relatively early stage of ReSPECT implementation, it is also possible that doctors had not yet made the conceptual shift from a DNACPR form to the more holistic approach of the ReSPECT process. Similarly, ReSPECT's key aim – to encourage a patient-centred approach to emergency care treatment planning by prompting patients' explicit involvement in

the discussion – was not often realised. This was exemplified by the finding that many of the participating consultants used ReSPECT conversations to inform patients or their relatives about a clinical decision, or to steer them toward a particular decision, rather than engage them in a more open-ended discussion of their wishes and preferences. Moreover, the consultants' focus on patients for whom treatment escalation was not medically indicated also means that other patients, for whom treatment escalation is medically indicated but who may wish to refuse these treatments, may not be given the opportunity to have their wishes respected. This suggests that, at early stages of implementation, the potential of ReSPECT to provide a more holistic patient-centred approach to decision making had not yet been realised fully" (p. 15, lines 603-620).

18. **P15, L 22-27** „Because the ReSPECT process requires clinicians to hold conversations with patients or their relatives before recording a decision about emergency care and treatments, it may encourage clinicians to articulate decision-making processes and to develop strategies for managing and communicating uncertainty“: This sentence is not derived from the results and thus is not an apt element of a conclusion; rather, it seems an appraisal of the tool that according to the COI Statement, 2 of the authors have developed. The following sentence, beginning with „These findings“, has no clear reference, since the quoted sentence does not refer to a finding, but praises the ReSPECT tool on general grounds. In contrast, while the ReSPECT form does include a section where the physician is expected to „provide clinical guidance on specific interventions that may or may not be wanted or clinically appropriate, including being taken or admitted to hospital +/- receiving life support“, I haven't found a single report in the study about a concrete statement referring to any other medical treatment option than CPR.

Response: We have modified the conclusion to include a more critical appraisal of the findings:

“While some consultants are using the ReSPECT process to broaden conversations about future emergency care treatment plans, many still focus on the decision regarding cardiopulmonary resuscitation and conversations often focus more on communicating and explaining clinical recommendations to patients and their families rather than exploring the patients' values and preferences to inform the decision. This suggests that the aims of the ReSPECT process are yet to be fully realised” (p. 16, lines 640-645).

VERSION 2 – REVIEW

REVIEWER	Caroline Nicholson University of Surrey, UK
REVIEW RETURNED	28-Sep-2019

GENERAL COMMENTS	Thank-you, the revisions to the paper strengthen an important contribution to the evidence - I look foreword to seeing it in print
--

REVIEWER	Jürgen in der Schmitten Institute of General Practice Medical Faculty Heinrich-Heine-University of Düsseldorf Germany
REVIEW RETURNED	30-Sep-2019

GENERAL COMMENTS	Thank you for your appreciation of the reviewers' comments, and for the revised manuscript. In my eyes really a beautiful paper, and I look forward to its publication. I have re-read the entire paper, and again very much enjoyed the results section. Now, I am also very happy with the discussion section, feeling that it does now very well relate to your findings, and gives valuable insights in the decision-making you observed. If anything, I would invite you to reevaluate the background's first sentence. We may have differing views on this, but in my eyes this first sentence is redundant to the second, and would not be missed if deleted. As it is, it somehow makes me uncomfortable: What should, given the outcome data on CPR, a clinician's reflection that "the burdens of cardiopulmonary resuscitation (CPR) for the patient outweigh potential benefits" be other than the prediction that CPR would not succeed (or a blunt paternalism that none of us would consider justifying)? My suggestion therefore is to begin the background with your second sentence: DNACPR decisions can be made when clinicians...
--

VERSION 2 – AUTHOR RESPONSE

Thank you for the positive and encouraging feedback.

We have made one additional change, in response to Reviewer 2's suggestion that we re-evaluate the introduction's first sentence. We now clarify that, in the UK context where the study was conducted, clinical practice guidelines include the weighing of burdens versus benefits in decision-making regarding CPR. We have combined the first and second sentences of the introduction, and revised the text as follows:

"UK clinical practice guidelines indicate that cardiopulmonary resuscitation (CPR) may be withheld when clinicians predict it would not succeed, if the patient refuses CPR, or following careful clinical assessment of the benefits and burdens of CPR" (p. 3, lines 50-52, in the manuscript without tracked changes).